# From real-time calibrations to smart HV tuning for FAIR

**Valentin Kladov**[1,2★]**, Johan Messchendorp**[2†] **and James Ritman**[1,2,3]

**1** Ruhr-Universität Bochum, Bochum, Germany
**2** GSI Helmholtzzentrum für Schwerionenforschung GmbH, Darmstadt, Germany
**3** Forschungszentrum Jülich GmbH, Jülich, Germany

★ V.Kladov@gsi.de , † J.Messchendorp@gsi.de

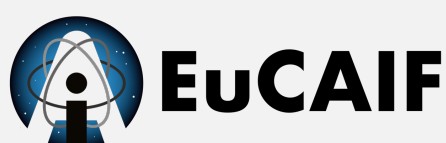

*The 2nd European AI for Fundamental Physics Conference (EuCAIFCon2025) Cagliari, Sardinia, 16-20 June 2025*

## Abstract

**Real-time data processing of the next generation of experiments at FAIR requires reliable event reconstruction and thus depends heavily on in-situ calibration procedures. Previously, we developed a neural-network-based approach that predicts calibration parameters from continuously available environmental and operational data and validated it on the HADES Multiwire Drift Chambers (MDCs), achieving fast predictions as accurate as offline calibrations. In this work, we introduce several methodological improvements that enhance both accuracy and the ability to adapt to new data. These include changes to the input features, better offline calibration and trainable normalizations. Furthermore, by combining beam-time and cosmic-ray datasets, we demonstrate that the learned dependencies can be transferred between very different data-taking scenarios. This enables the network not only to provide real-time calibration predictions, but also to infer optimal high-voltage settings, thus establishing a practical framework for a real-time detector control during data acquisition process.**

# 1 Introduction

Future FAIR experiments such as CBM [1] and PANDA [2] will operate at interaction rates several orders of magnitude above present facilities at GSI, such as HADES [3], requiring high level triggering and online event reconstruction to keep data volumes manageable [4]. Calibrations will also have to be performed during data taking. For tracking detectors, frequent calibrations are especially needed to maintain tracking and particle-identification performance. Since traditional procedures for almost any calibration rely on full tracks reconstruction, they are computationally too demanding for real-time operation [5].

An alternative is to exploit environmental and operational parameters, such as pressure, gas composition, or high voltage, to predict detector's performance, i.e. calibration constants, without processing physics events. Following this concept, we developed a neural-network-based prediction tool using data from the HADES experiment. While HADES does not use online reconstruction, its gaseous Multiwire Drift Chambers (MDCs) provide an excellent test case, as their gain strongly depends on environmental conditions and requires continuous monitoring with parameters measured and stored.

The setup of the detector and the neural network training procedure were described in our previous report [6]. In this work, we discuss testing the network with different beam times and cosmic data, comparing several architectures, and exploring the possibility of smart high-voltage (HV) tuning enabled by the trained model. For this purpose, several upgrades were introduced.

Breaks between the annual HADES beam times often involve detector improvements that significantly alter both the operational settings, used as network input, and the calibration values used as target. The method was based on data taken in February 2022, and is now tested with more recent data taken in April 2025. Between these periods, the MDC front-end electronics was substantially upgraded. To accommodate these changes, we implemented a trainable exponential normalization of both input and output values inside the model, preceded by rescaling the inputs to a fixed range (e.g. 1–10). This procedure allows the network to handle parameters with widely different means and standard deviations in a consistent way.

Another important upgrade concerns input and target values. In addition to the previously used parameters, the input now includes gas concentrations ($H_2O$, $N_2$) and detector count rates. While these parameters are not strictly necessary for a proof-of-principle, they improve stability by introducing higher-order dependencies. Regarding the target values, time-over-threshold distributions are now fitted in a more stable algorithm with a Landau–Gaussian convolution (Langaus), which provides a more accurate description of ionization-loss distributions than a pure Landau fit.

Finally, for testing smart HV tuning, an additional issue had to be addressed. The February 2022 beam time was conducted at nearly constant HV, which is a crucial input variable for predictions and correlations. As a result, the network trained on these data became insensitive to HV variations. In the view of this, we introduced an additional dense layer in the middle of the network that directly combines the HV input with intermediate features, thereby preserving gradients and retaining sensitivity to HV adjustments. The details of incorporating cross-dataset dependencies are discussed in the next section.

# 2 Results

In our previous work [6] we demonstrated the feasibility of the approach, producing very fast predictions with accuracy comparable to the offline calibration used for training. Here, to compare different models and evaluate performance quantitatively, we use the following

metrics: Root Mean Square Error (RMSE), where each entry is normalized by the target error; Precision, which measures the shifts between adjacent predictions relative to the shifts of the target values and directly reflects the level of overfitting; Robustness, defined as the increase in RMSE on the test dataset; and Latency, i.e. the time required for a single forward pass.

With the recently developed methods and upgrades, it was possible to train various architectures to yield similar performance, while previously a complex graph convolutional network was needed. Typical values achieved for all models are the following: RMSE $\approx 1.5\sigma$, precision $\approx 60\%$, and robustness $\approx 10\%$ for a period of five days after the training dataset ended. These results indicate that the network does not overfit even without explicit L2 regularization, although the predictions are still somewhat less accurate than the offline calibration. However, a closer inspection suggests that the errors of the offline calibration itself are likely underestimated. Within this framework, we compared different fully connected networks, two versions of graph-convolutional LSTMs (spatial and Chebyshev convolutions) and transformers. The transformer model performed worst due to the complexity of the attention mechanism and the associated difficulties in hyperparameter tuning.

The learned dependencies can be used both for the fast prediction of calibration constants and for optimizing detector settings to achieve more stable performance, for example, constant calibration values. Since the input features include detector settings such as high voltage (HV), the network can be reversed to infer optimal HV values given the remaining inputs and a desired calibration factor. To address the fact that HADES beam time data were taken with nearly constant HV, we acquired additional cosmic-ray data with manually varied settings. Using these data, together with the new approaches of trainable normalization and smart HV propagation within the network, we fine-tuned the model pre-trained on beam-time data. During fine-tuning, we froze the majority of the network weights that stand before the inserted HV layer, ensuring that only the corresponding dependencies were adjusted.

After the learned HV-related weights were frozen, the network was applied back to the beam-time data with artificially modified HV values. The resulting dependencies from fine-tuning on cosmic data and testing on beam time data are shown in Fig. 1. The two sets of correlations are in good agreement, demonstrating the possibility of merging significantly different datasets, where one parameter mostly fluctuates in one dataset and others mostly in another. At this stage, reversing the network to infer HV settings is straightforward. In real-time operation, with automatically tuned HV values, the required dataset will be constructed automatically, further improving the quality of the learned cross-dependencies with high voltage.

## 3 Conclusion and Outlook

A neural-network-based method for predicting ionization-loss calibration parameters of the HADES MDC has been developed and validated. In this work, the approach was further improved by introducing an expanded set of environmental input features, a more accurate offline calibration using Langaus fits, and a trainable normalization scheme that stabilizes training across different detector conditions. In addition, the treatment of high-voltage input was refined, enabling the network to capture HV dependencies just from fine-tuning.

By combining beam-time and cosmic-ray datasets, where different parameters fluctuate, we demonstrated that the model can reliably transfer learned dependencies and be inverted to determine optimal HV settings. The resulting tool provides both accurate calibration predictions and a practical framework for smart high-voltage tuning in gaseous tracking detectors.

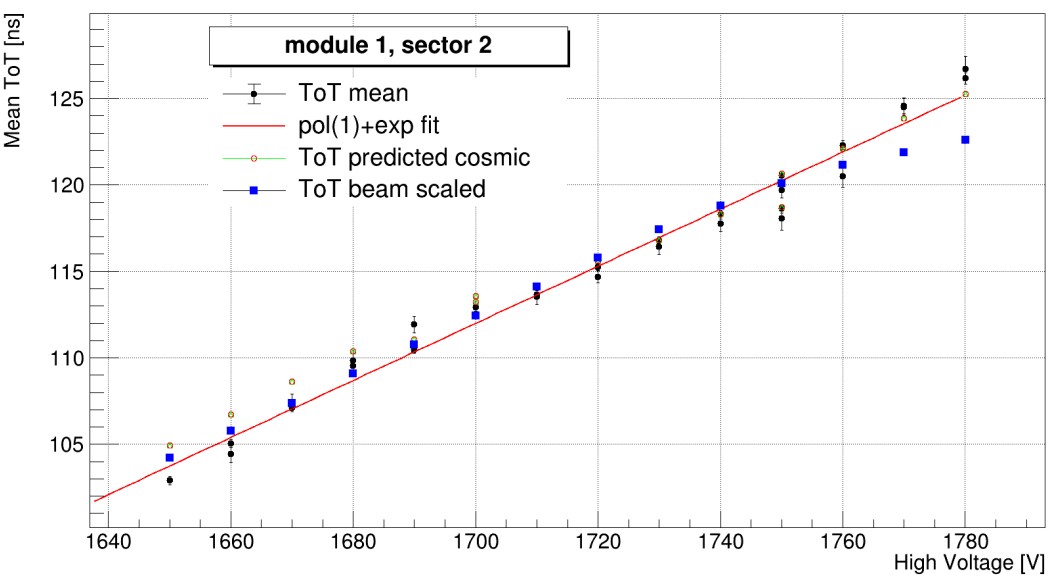

Figure 1: Average Time-over-Threshold (ToT) as a function of high voltage for one MDC chamber. Shown are offline calibration results from cosmic data (black points), network predictions after fine-tuning on cosmic data (red circles), and beam-time data with artificially varied HV values (blue squares). The agreement between tests demonstrates that HV dependencies can be transferred between datasets.

## Acknowledgements

This work was supported by the Ministry of Culture and Science of the State of Northrhine Westphalia (Netzwerke 2021, NRW-FAIR), as well as by the HADES collaboration and the GSI computing cluster.

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
