# Peer review of "From real-time calibrations to smart HV tuning for FAIR"

_SciPost Physics Proceedings_

## Round 1 · Referee Report · Anonymous (Referee 1) · 2025-11-23

Report

Congratulations to the authors on this interesting and well-written article. The problem of developing accurate in situ calibration is of great importance, and this work demonstrates a very good command of the tools needed to implement this procedure.

My main regret in reading this article is that it is not sufficiently explicit and lacks the details and explanations necessary for a better understanding. Even reading reference [6] did not help me grasp the techniques and methods presented. I understand that this is a proceeding and that it is limited in terms of page count, but I wonder to what extent it may be of interest to people who are not closely involved in the issues encountered at FAIR experiments. A detailed publication on this work would be certainly be welcome at some point.

Here are a few comments, along this line of thinking. I apologize if they may sound negative, but it's really because I was rather limited in my understanding when reading this article.

Comments:
• Give full acronyms for FAIR, CBM and PANDA
• avoid qualitative comments, such as in 1st line "several ordrer of magnitude"
• Please give at least a short description of the NN architecture described in [6]
• I have no idea what is "smart high voltage". Please explain
• Why is a trainable exponential normalization needed ? Why this specific fixed range ?
• What do you mean by "higher order dependencies" ? How can you convince the reader that this is indeed the case ?
• About RMSE, you write "where each entry is normalized...". which entries ? All ? Only output values ? Please clarify.
• The section "Using this data" to "dependencies were adjusted" is very cryptic to me. Please help anyone who isn't working on this topic understand it.

Recommendation

Ask for minor revision

---

## Round 2 · Author Response

List of changes
point-by-point responses, arranged according to the comments:
-
“Give full acronyms for FAIR, CBM and PANDA.” – FAIR → Facility for Antiproton and Ion Research – CBM → Compressed Baryonic Matter experiment – PANDA → anti-Proton ANnihilation at DArmstadt experiment
-
“Avoid qualitative comments, such as in 1st line ‘several order of magnitude’.” – several orders of magnitude → 2–3 orders of magnitude (comparing up to 50kHz at HADES to Planned 10MHz at CBM).
-
“Please give at least a short description of the NN architecture described in [6].” Added a concise architectural description, including: – Input format, – Use of graph-convolutional LSTM and FCN (with added citations), – Output tensor and training idea.
-
“I have no idea what is ‘smart high voltage’. Please explain.” Removed the ambiguous phrasing and replaced it with a clear description of the method: – Introduced the concept as 'autonomous real-time runing of the HV'.
-
“Why is a trainable exponential normalization needed? Why this specific fixed range?” – Added an explicit mathematical definition of the normalization where only the width scales exponentially – Added justification for rescaling to a fixed range (1–10) and trainable normalization, starting with "To allow the network to accommodate such changes..."
-
“What do you mean by ‘higher-order dependencies’? How can you convince the reader that this is indeed the case?” Clarified the concept: – Explained that additional environmental variables enable second-order dependencies in a particular dataset (not physics-wise, derived experimentally)
-
“About RMSE, you write ‘where each entry is normalized…’. Which entries? All? Only output values? Please clarify.” Rewrote the RMSE definition: – Differences are normalized by target errors so that the RMSE is non-dimensional, – Averaging is performed over all entries in the dataset (mathematical procedure is the same for any dataset, entries are time periods called 'runs' internally).
-
“The section ‘Using this data’ to ‘dependencies were adjusted’ is cryptic. Please clarify.” Rewrote this part for clarity starting with 'Using the new approaches...': – Explained the fine-tuning procedure, – Highlighted which weights are frozen, – Clarified how only HV-related dependencies are made trainable during fine-tuning,
-
Added a few additional changes for clarity in connection to the comments: – For example, starting with 'With the recently developed methods and upgrades...' changed the wording for the comparison with GConvLSTM, as it is now introduced before (point 3.) – Added 'adjacent in time' in Precision definition, as now 'adjacent' may be understood ambiguously

---

## Round 2 · List of Changes

point-by-point responses, arranged according to the comments:
-
“Give full acronyms for FAIR, CBM and PANDA.” – FAIR → Facility for Antiproton and Ion Research – CBM → Compressed Baryonic Matter experiment – PANDA → anti-Proton ANnihilation at DArmstadt experiment
-
“Avoid qualitative comments, such as in 1st line ‘several order of magnitude’.” – several orders of magnitude → 2–3 orders of magnitude (comparing up to 50kHz at HADES to Planned 10MHz at CBM).
-
“Please give at least a short description of the NN architecture described in [6].” Added a concise architectural description, including: – Input format, – Use of graph-convolutional LSTM and FCN (with added citations), – Output tensor and training idea.
-
“I have no idea what is ‘smart high voltage’. Please explain.” Removed the ambiguous phrasing and replaced it with a clear description of the method: – Introduced the concept as 'autonomous real-time runing of the HV'.
-
“Why is a trainable exponential normalization needed? Why this specific fixed range?” – Added an explicit mathematical definition of the normalization where only the width scales exponentially – Added justification for rescaling to a fixed range (1–10) and trainable normalization, starting with "To allow the network to accommodate such changes..."
-
“What do you mean by ‘higher-order dependencies’? How can you convince the reader that this is indeed the case?” Clarified the concept: – Explained that additional environmental variables enable second-order dependencies in a particular dataset (not physics-wise, derived experimentally)
-
“About RMSE, you write ‘where each entry is normalized…’. Which entries? All? Only output values? Please clarify.” Rewrote the RMSE definition: – Differences are normalized by target errors so that the RMSE is non-dimensional, – Averaging is performed over all entries in the dataset (mathematical procedure is the same for any dataset, entries are time periods called 'runs' internally).
-
“The section ‘Using this data’ to ‘dependencies were adjusted’ is cryptic. Please clarify.” Rewrote this part for clarity starting with 'Using the new approaches...': – Explained the fine-tuning procedure, – Highlighted which weights are frozen, – Clarified how only HV-related dependencies are made trainable during fine-tuning,
-
Added a few additional changes for clarity in connection to the comments: – For example, starting with 'With the recently developed methods and upgrades...' changed the wording for the comparison with GConvLSTM, as it is now introduced before (point 3.) – Added 'adjacent in time' in Precision definition, as now 'adjacent' may be understood ambiguously

---

## Editorial Decision

unknown